# Autonomous Penetration Testing Based on Improved Deep Q-Network

**Shicheng Zhou** [1,2], **Jingju Liu** [1,2,*], **Dongdong Hou** [1,2], **Xiaofeng Zhong** [1,2] **and Yue Zhang** [1,2]

1   College of Electronic Engineering, National University of Defense Technology, Hefei 230000, China; zhoushicheng@nudt.edu.cn (S.Z.); houdong1992@gmail.com (D.H.); zhongxiaofeng17@nudt.edu.cn (X.Z.); zhangyue@nudt.edu.cn (Y.Z.)
2   Anhui Province Key Laboratory of Cyberspace Security Situation Awareness and Evaluation, Hefei 230037, China
*   Correspondence: jingjul@aliyun.com

**Abstract:** Penetration testing is an effective way to test and evaluate cybersecurity by simulating a cyberattack. However, the traditional methods deeply rely on domain expert knowledge, which requires prohibitive labor and time costs. Autonomous penetration testing is a more efficient and intelligent way to solve this problem. In this paper, we model penetration testing as a Markov decision process problem and use reinforcement learning technology for autonomous penetration testing in large scale networks. We propose an improved deep Q-network (DQN) named NDSPI-DQN to address the sparse reward problem and large action space problem in large-scale scenarios. First, we reasonably integrate five extensions to DQN, including noisy nets, soft Q-learning, dueling architectures, prioritized experience replay, and intrinsic curiosity model to improve the exploration efficiency. Second, we decouple the action and split the estimators of the neural network to calculate two elements of action separately, so as to decrease the action space. Finally, the performance of algorithms is investigated in a range of scenarios. The experiment results demonstrate that our methods have better convergence and scaling performance.

**Keywords:** penetration testing; reinforcement learning; cybersecurity; DQN algorithm

## 1. Introduction

Penetration testing (short PT or pentesting) is active and authorized simulated cyberattack, aiming at assessing cybersecurity and discovering the hidden vulnerabilities. Currently, pentesting plays a crucial role in strengthening the defense of computer systems against cyberattacks, as digital assets are more frequently exposed to hackers' persistent, varied, and complex threats than ever before.

However, the traditional pentesting methods mainly rely on highly skilled cybersecurity experts with domain-specific knowledge and experience, which requires prohibitive labor and time costs. Consequently, pentesting is conducted infrequently in many organizations. Autonomous pentesting can be a solution to this problem. Compared to the human-based method, performing pentesting autonomously is a more efficient and intelligent way. It can realize regular security testing without expensive specialists and make the pentesting process accessible to those nonexperts.

Researches on autonomous pentesting originate from the attack tree [1] and attack graph [2]. Both methods offer interpretable and formal models for evaluating system security and discovering potential attack paths. The limitation of them is that they need complete information of the target topology and all hosts' configuration, which is unrealistic from the perspective of real-world attackers. They cannot be applied to large network scenarios due to the state space explosion and their complex modeling process. Another approach to autonomous pentesting is the partially observable Markov decision process (POMDP) model [3–6]. Using the POMDP method allows modeling the incomplete

knowledge characteristic of the real-world hackers and accounting for the uncertainty in the pentesting process. However, the POMDP-based method is computationally expensive in large-scale network scenarios.

Currently, remarkable success has been achieved by reinforcement learning (RL), and in some games, the RL-based agents have outperformed human players, such as AlphaGo [7], OpenAI Five [8], Alpha Star [9], and MOBA AI [10]. Similar to many game rules, pentesting is also a process of dynamic decision making based on the state of the environment. Expressing the pentesting process in terms of a game is a promising way to realize autonomous pentesting. The RL-based pentesting agent can be trained to observe the network environment and learn the optimal policy using the trial-and-error method [11]. One of the main advantages of using the reinforcement learning approach is that the agent can learn without assumed prior knowledge for any given scenario.

Recently reinforcement learning algorithms, such as tabular Q-learning and deep Q-network (DQN), have been applied to solve the pentesting problem in some researches. Tabular Q-learning has been applied to solving CTF (capture the flag) problems [12] and exploiting SQL injection vulnerabilities [13], proving the feasibility of applying reinforcement learning to simple and particular autonomous pentesting problems. Compared to the tabular Q-learning, DQN based methods [14–16] have a natural advantage in dealing with large state pace problems using the deep neural network for function approximation. However, the previous DQN-based methods are still troubled in the sparse reward problem [17] and the large action space caused by the increase in network size. Current research studies on autonomous pentesting based on reinforcement learning have only been applied in simple network scenarios, and there is a large room for improvement in convergence and scaling performance.

On the basis of previous research studies, this paper models pentesting as a Markov decision process (MDP) problem and analyzes the main challenges of autonomous pentesting in large-scale networks. Then, an improved DQN algorithm is proposed to train the pentesting agent that can learn the optimal attack policy without prior knowledge. The proposed algorithm has better convergence performance in large-scale networks. Specifically, five improvements of the DQN algorithm are selected reasonably and combined together to alleviate the sparse reward problem. Then, in order to decrease the large action space, we decompose the estimator of DQN to separately output the two elements of the attack vector. In the end, experiments are implemented with NASim [18], which is an abstract and simulated experimentation research platform that can be used as the benchmark to test the RL-based pentesting agents.

The rest of this paper is organized as follows. In Section 2, we introduce the background of autonomous penetration testing, including the concept and general process of pentesting and the main challenges of using RL for autonomous pentesting in large-scale networks. In Section 3, we describe the detailed improvement methods of the proposed algorithm for the current challenges. Subsequently, in Section 4, we demonstrate the results and discussion of our experiments, and in Section 5, we draw conclusions and present future work.

## 2. Background

### 2.1. Penetration Testing

Penetration testing is a network security testing and evaluation method that aims to identify the vulnerabilities in the computer system and discover the possible attack path of the hackers. MITRE matrix [19] provides a series of tactics and techniques for pentesting. The process of pentesting can be simply summarized as the information gathering phase, access gaining phase, and trace erasing phase. Specifically, attackers have to use scanning tools to further their knowledge of the target. The information they gather covers the operating system (OS), running services, and other vulnerability-relevant information. Then, based on the information gathered before, they use the payload to exploit the discovered vulnerability in the target system with the aim of gaining control

access. However, the access they gained is usually limited. Attackers sometimes need privilege escalation to make sure that they can collect as much sensitive data as possible. Finally, hackers in the real world have to clear their attack traces to remain anonymous.

### 2.2. Markov Decision Process and Reinforcement Learning

Markov decision process means to be a mathematical framework for describing multistage decision making (or sequential decision) problems [20], and reinforcement learning is a technique for solving MDPs problems. An MDP problem can be formally defined by the tuple: $< \mathcal{S}, \mathcal{A}, \mathcal{R}, \mathcal{T} >$ , where $\mathcal{S}$ is the state space representing all the possible state of the environment, $\mathcal{A}$ is the action space representing all the possible actions of the agent, $\mathcal{R}$ is reward function representing the immediate reward after performing an action, $\mathcal{T}$ is the transition function $\mathcal{T}(s, a, s') = P(s'|s, a)$ representing the probability of environmental state transition after taking an action.

Different from supervised learning and unsupervised learning, reinforcement learning learns the mapping from state to action through continuous interaction with the uncertain environment [21]. More specifically, the agent observes the environment and receives the state $s_t \in \mathcal{S}$ at time $t$; then, the action $a_t \in \mathcal{A}(s_t)$ is selected based on the policy $\pi$. As the result of conducting the selected action, the agent can obtain the reward $r_t \in \mathcal{R}$ from the environment; meanwhile, the environment state transitions to $s_{t+1}$. The process of the agent interacting with the environment from $s_t$ to $s_{t+1}$ is called a step, and the collection of steps from the initial step to the end is called an episode. The goal of the agent is to learn the optimal policy $\pi^*$ that maximizes the cumulative rewards or average rewards per episode. The policy is the mapping from state to action, and the optimal policy $\pi^*$ can be expressed as the formula:

$$\pi^* = argmax_\pi \sum_{t=0}^{T} \gamma^t \mathbb{E}_\pi[r_t], \tag{1}$$

where $\gamma \in (0, 1)$ is a parameter named discount factor measuring the importance of present rewards to future rewards. The smaller the value, the more shortsighted the agent will be, and on the contrary, it will pay more attention to future long-term rewards.

There are many reinforcement learning algorithms available for solving MDP problems. Q-learning is a model-free and value-based algorithm that is widely used. Model-free means it attempts to learn the optimal policy by trial-and-error experience without the need of modeling the environment. Value-based means the policy is derived directly from the value function. For the Q-learning algorithm, the policy can be extracted by the Q-function $Q(s_t, a_t)$ that represents the expected reward if the agent conducts the action $a_t$ from the state $s_t$. The Q-function will converge with the exploration and exploitation of the environment, and then the agent can use it to choose the greedy action for a state. The update equation of the Q-function is:

$$Q^{NEW}(s_t, a_t) \leftarrow Q(s_t, a_t) + \alpha(r_t + \gamma max_a(s_{t+1}, a_t) - Q(s_t, a_t)), \tag{2}$$

where $\alpha \in (0, 1]$ is the learning rate and $\gamma$ is the discount factor.

The traditional tabular Q-learning algorithm uses a table to store the state-action pair values during the training process. This method is simple to implement, though it cannot be applied to solve the problem with large state space as the table size is limited. Deep reinforcement learning (DRL) is the combination of deep learning and reinforcement learning, which makes it possible to cope with the curse of dimensionality. Minh et al. [22,23] proposed the DQN algorithm that uses the deep neural network as the function approximator to generate the state-action value. In order to ensure the algorithm can converge, DQN uses the experience replay mechanism to improve the efficiency of the use of previous experience and remove the correlation between data by taking random samples from the memory. To improve the stability of the algorithm, DQN maintains two separate networks (the $Q$ network with weights $\theta$ and the target $Q$ network with weights $\theta^-$) to generate the predicted $Q$ value and target $Q$ value. The target $Q$ network keeps frozen for certain steps;

then, the weights of the $Q$ network will be assigned to the target $Q$ network. The target $Q$ value $y_i$ at $i$ iteration is calculated as Equation (3), and the loss function of DQN is defined as Equation (4).

$$y_i = r + \gamma max_{a'} Q(s', a'; \theta_i^-), \tag{3}$$

$$L_i(\theta_i) = \mathbb{E}_{\pi \theta_i}[(y_i - Q(s, a; \theta_i))^2], \tag{4}$$

In recent years, many researchers have made improvements to the DQN algorithm from different perspectives, and we will review those improved versions in Section 3 in a detailed manner.

### 2.3. Modeling Pentesting as an MDP Problem

To use RL for autonomous pentesting, we model pentesting as an MDP problem that is defined by the tuple $< \mathcal{S}, \mathcal{A}, \mathcal{R}, \mathcal{T} >$. In the autonomous pentesting task, the state refers to the information observed by the agent on the network environment, including the configure knowledge of all hosts and vulnerability information, etc. The size of the state space is affected by the number of hosts in the network, which is usually exponential [15]. The action refers to the vulnerability exploitation, scanning, and privilege escalation operations performed by the agent on the target host. An action $a$ can be seen as an attack vector defined by $< h, o >$, which means the agent takes operation $o$ on the host $h$. At any training step, the size of the agent's action space reaches $O(M \times N)$, where $M$ is the number of hosts in the network and $N$ is the number of executable operations for attacking the target. Reward is the driver of the agent. We define the reward function as the value of all compromised hosts minus the cost of all actions:

$$\mathcal{R} = \sum_{h \in H} value(h) - \sum_{a \in \mathcal{A}} cost(a), \tag{5}$$

where $H$ represents the set of compromised hosts, $\mathcal{A}$ represents the set of actions the agent takes, $value(h)$ measures the importance of any hosts and returns the value of compromised host $h$, and $cost(a)$ returns the cost of action $a$. Based on the reward function $\mathcal{R}$, the goal of the agent is to compromise the most valuable hosts with the least costly actions as much as possible, which mimics the hackers in the real world. The transition function certainly keeps unknown for the reason that we attempt to use the model-free algorithm.

The agent is expected to complete the pentesting process autonomously and learn the optimal attack path that has maximized rewards. The attack path is the sequence of actions or a series of ordered attack vectors, and the search space of attack paths is affected by the network scale, thereby affecting the learning efficiency of the agent. Specifically, when the network size increases, the learning efficiency of the agent is mainly limited by two challenges:

1.  The sparse reward problem. Rewards work as intermediate feedback on progress towards the goal, without which learning becomes impossible [24]. However, there exist only a few sensitive hosts with positive value in the network usually, making rewards become sparse when the network size increases. Thus, the algorithm becomes difficult to converge.
2.  Large action space problem. The agent learns optimal policy through trial and error. However, at each training step, an action is determined by two dimensions: the agent not only needs to choose the attack target but also needs to choose the proper operation. Action space increases with network size, which reduces exploration efficiency and increases the trial-and-error cost.

## 3. Methodology

In this section, we introduce our methods for handling the two problems mentioned above. To alleviate the sparse reward problem, we propose an algorithm named NDSPI-DQN that integrates five extensions to DQN: the N oisy nets, Dueling network architectures, Soft Q-learning, Prioritized experience replay, and Intrinsic curiosity module. Furthermore,

to reduce the action space, we decouple the attack vector and split the estimator of DQN to calculate two elements of action separately.

### 3.1. Extensions to DQN

Five extensions to DQN are selected for the reason that they are proved to strengthen the exploration of the agent from different perspectives and each of them improves the overall performance.

Soft $Q$-learning [25] is a method proposed for learning the maximum entropy policies. It allows for learning diverse strategies and better exploration, and it is proved to perform better in continuous action tasks. When combining it with DQN to handle discrete action tasks, the target $Q$ value is calculated as Equation (6), where $\sigma$ is a parameter used to measure the relative importance of entropy and reward.

$$y^{soft} = r + \gamma[\sigma log \sum_{a'} exp(\frac{1}{\sigma}Q(s',a';\theta^-))]. \tag{6}$$

As for the action selection strategy, instead of using the greedy policy in traditional DQN, soft Q-learning uses a *softmax* policy as Equation (7), which improves the exploration by assigning a probability of being selected to each action.

$$\pi(a|s) = softmax_a(\frac{1}{\sigma}Q(s,a;\theta)). \tag{7}$$

Dueling network [26] improves the performance of policy evaluation by changing the neural network architecture of DQN. It decomposes the $Q$ value function as the sum of value function and advantage function:

$$Q(s,a;\psi) = V(s;\theta,\beta) + [A(s,a;\theta,\alpha) - \frac{1}{|\mathcal{A}|}\sum_{a'} A(s,a';\theta,\alpha)], \tag{8}$$

where $\theta$, $\alpha$, and $\beta$ denote respectively the shared network parameters, the value stream parameters, and the advantage stream parameters, and $\psi$ is their concatenation.

DQN uses experience replay to ensure that samples are independent and identically distributed, however, the traditional random sampling method ignores the significance of different samples. In order to make experience replay more efficient, prioritized experience replay [27] uses temporal difference error (TD error, $\delta$) to measure the significance of transitions. It samples transition with probability

$$P(i) = \frac{p_i^\alpha}{\sum_k p_k^\alpha}, \tag{9}$$

where $\alpha$ denotes how much prioritization is used and $p(i) = |\delta_i| + \mu$ is the priority of transitions with a small positive constant $\mu$.

Noisy net [28] improves efficient exploration by adding parametric Gaussian noise to the linear layer of DQN. DQN applies $\epsilon$-greedy policy to select actions greedily based on the $Q$ function or randomly pick actions with probability $\epsilon$, compared to which noisy net has more abundant exploration on problems with large action space. The noisy linear layer of noisy net is defined as

$$y = b + b_{noisy} \odot \varepsilon^b + (W + W_{noisy} \odot \varepsilon^w)x, \tag{10}$$

where $\varepsilon^b, \varepsilon^w$ denote the noise random variables, and $\odot$ represents the elementwise product. This transformation is used to replace the standard linear $y = b + Wx$.

Pathak et al. [29] use curiosity as the internal reward signal to drive the agent to explore in an environment where external rewards are sparse. The external reward is given by the environment while the internal is generated from the self-supervised intrinsic

curiosity module (ICM); thus, the goal of the curiosity-driven agent is maximizing the sum of internal rewards $r^i$ and external rewards $r^e$. ICM consists of two submodels: the inverse model is trained to predict the action $\hat{a}_t$, given the feature vectors $\phi(s_t), \phi(s_{t+1})$ that are encoded by the state $s_t, s_{t+1}$; the forward model takes as input $\phi(s_t), a_t$ and predicts the feature vector of the next state $\hat{\phi}(s_{t+1})$. Curiosity is defined as the prediction error of the environment state; thus, the internal reward is calculated as

$$r_t^i = \frac{\eta}{2} \left\| \hat{\phi}(s_{t+1}) - \phi(s_{t+1}) \right\|_2^2, \tag{11}$$

where $\eta > 0$ is a scaling factor. The inverse and the forward model are respectively optimized by minimizing the loss function $L_I(a_t, \hat{a}_t)$ and $L_F(\phi(s_{t+1}), \hat{\phi}(s_{t+1}))$; then, the overall loss function of ICM is the weighted sum of them.

*3.2. Integration and Decoupling*

We combine those extensions together and propose the NDSPI-DQN algorithm to train the pentesting agent. As shown in Figure 1, we apply the dueling network architecture with added noise for both the $Q$ network and the target network. The input of the neural network is the environment state vector, the hidden layers are fully connected to the input and output layers, and the output is the predicted values of all actions. $\epsilon$-greedy policy is no longer used, but the action is directly selected by the *softmax* policy as Equation (7). The goal is set to maximize both the external reward and internal reward that is generated by ICM. Prioritized experience replay to store and sample important transitions more frequently. After combining those extensions, the action-value function can be written as $Q(s, a, \varepsilon; \psi)$, where $\varepsilon$ and $\psi$, respectively, denote the noise variable and the concatenation network parameters. Thus, the target $Q$ value is calculated as

$$y = r + \gamma[\sigma log \sum_{a'} exp(\frac{1}{\sigma} Q(s', a', \varepsilon'; \psi^-))], \tag{12}$$

where $r = r^i + r^e$ and the overall loss function is defined as

$$L(\psi, \theta_I, \theta_F) = \lambda \mathbb{E}_{(s,a,r,s') \sim D}[y - Q(s, a, \varepsilon; \psi)]^2 + (1 - \delta)L_I + \delta L_F, \tag{13}$$

where $\lambda$ and $\delta$ are the scalar that weights the importance of the loss of the policy network and ICM model.

However, it is not enough to merely improve the agent's exploration. In order to reduce the cost of trial and error in the exploration process of the agent, we reduce the action space by decoupling the action that can be seen as an attack vector $a = <h, o>$. As illustrated in Figure 2, the crucial insight of our method is that we split the the NDSPI-DQN network into two separate streams: one estimates the value of hosts, and the other estimates the value of the operations, which means the agent can independently select the victim host and the operation taken against the target at one time according to the current environment state vector. For the agent in an environment with $M$ target hosts and $N$ executable operations, the output of the neural network is divided into two parts: the $Q$ values of $M$ hosts and the $Q$ values of $N$ operations; thus, the action space is reduced from the original $O(M \times N)$ to $O(M + N)$.

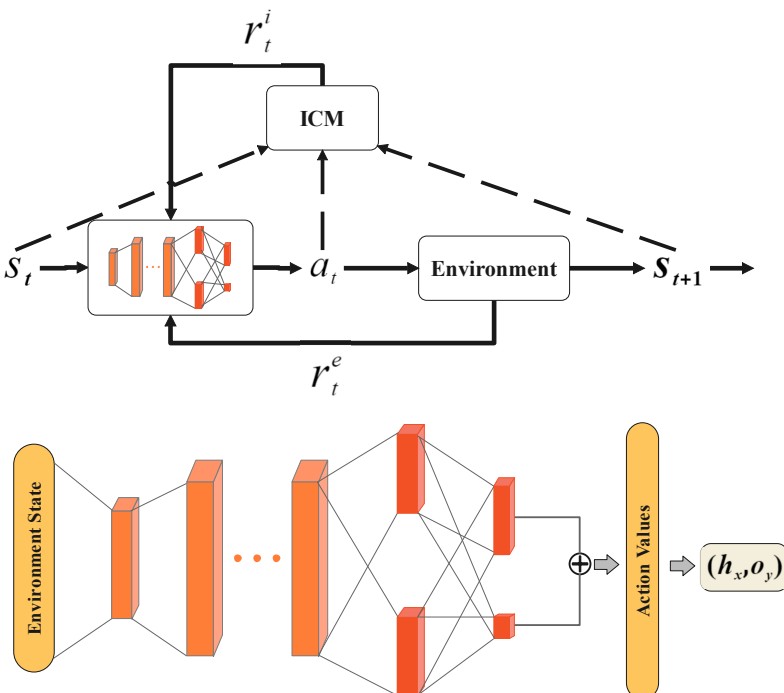

**Figure 1.** The learning cycle (**Top**) and schematic illustration of the NDSPI-DQN (bottom). (**Top**) The agent observes current environment state $s_t$ and outputs an action $a_t$ according to the learned policy; then, the external reward $r_t^e$ is obtained from the environment. Meanwhile, ICM takes $(s_t, a_t, s_{t+1})$ as input and outputs the internal reward $r_t^i$. Both $r_t^i$ and $r_t^e$ are used to optimize the policy. (**Bottom**) The details of the algorithm architecture are explained in Section 3.2. The orange layers denote fully connected layers and the red layers denote dueling network architectures with noisy linear layers. The input to the neural network is the vector of environment states, and the output is the action value.

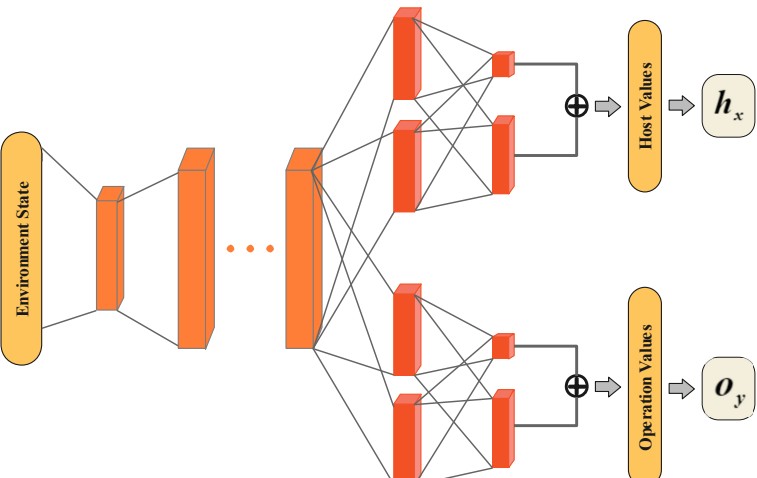

**Figure 2.** Schematic illustration of the decoupling NDSPI-DQN. Two streams share the hidden layers and each of them outputs its $Q$ value using the dueling network architecture with the noisy linear layers.

Two streams independently output the elements of actions given the input state using $\pi(h|s) = softmax_h(\frac{1}{\sigma_h}Q(s, h, \varepsilon; \psi_h))$ and $\pi(o|s) = softmax_o(\frac{1}{\sigma_o}Q(s, o, \varepsilon; \psi_o))$. The selected host $h$ and operation $o$ are then combined into the attack vector to act on the

environment. We use the average of the host $Q$ value and the operation $Q$ value to calculate the overall loss:

$$L(\psi, \theta_I, \theta_F) = \lambda \mathbb{E}_{(s,h,o,r,s') \sim D}[y - \frac{1}{2}(Q(s,h,\varepsilon;\psi_h) + Q(s,o,\varepsilon;\psi_o))]^2 + (1-\delta)L_I + \delta L_F, \quad (14)$$

where $\psi_h, \psi_o$ are the parameters of the host stream and operation stream, $\psi = \{\psi_h, \psi_o\}$ is the common network parameters and

$$y = r + \frac{\gamma}{2}\left[\sigma_h log \sum_{h'} exp(\frac{1}{\sigma_h}Q(s',h',\varepsilon';\psi_h{}^-)) + \sigma_o log \sum_{o'} exp(\frac{1}{\sigma_o}Q(s',o',\varepsilon';\psi_o{}^-))\right]. \quad (15)$$

The algorithm of NDSPI-DQN and its decoupling version are presented in Algorithm 1.

---

**Algorithm 1** NDSPI-DQN.

---

Initialize the replay memory $D$, the set of random variables $\xi$
Initialize the $Q$ network with weights $\psi$, the target $Q$ network with weights $\psi^-$
Initialize the ICM with weights $\theta_I, \theta_F$
Initialize the environment *env*, the boolean Decoupling: *False* for NDSPI-DQN and *True* for decoupling NDSPI-DQN
**for** episode = 1, M **do**
    Reset the environment state $s_0 \sim env$
    **for** step = 1, T **do**
        Set $s_t \leftarrow s_0$
        Sample the noisy network $\varepsilon \sim \xi$
        **if** Decoupling **then**
            Select $h_t \leftarrow softmax_h(\frac{1}{\sigma_h}Q(s_t,h,\varepsilon;\psi_h))$ , $o_t \leftarrow softmax_o(\frac{1}{\sigma_o}Q(s_t,o,\varepsilon;\psi_o))$
            Execute action $a_t = < h_t, o_t >$, obtain the external reward $r_t^e$ and next state $s_{t+1}$
            Store the transition $(s_t, h_t, o_t, r_t^e, s_{t+1})$ in $D$, update the priority of transitions
            Sample a minibatch of transitions $(s_j, h_j, o_j, r_j^e, s_{j+1})$ from $D$ with probability using Equation (9)
        **else**
            Select action $a_t \leftarrow softmax_a(\frac{1}{\sigma}Q(s_t,a,\varepsilon;\psi))$
            Execute action $a_t$, obtain the external reward $r_t^e$ and observer next state $s_{t+1}$
            Store the transition $(s_t, a_t, r_t^e, s_{t+1})$ in $D$ and update the priority of transitions
            Sample a minibatch of transitions $(s_j, a_j, r_j^e, s_{j+1})$ from $D$ with probability using Equation (9)
        **end if**
        Sample the noisy network $\varepsilon' \sim \xi$
        Calculate the loss functions $L_I, L_F$ of ICM and the internal reward $r_j^i$
        **if** episode terminate at step $j+1$ **then**
            Set $y_j = r_j^e + r_j^i$
        **else**
            **if** Decoupling **then**
                Set $y_j$ using Equation (15)
                Do a gradient step with the overall loss using Equation (14) with respect to $\psi$
            **else**
                Set $y_j$ using Equation (12)
                Do a gradient step with the overall loss using Equation (13) with respect to $\psi$
            **end if**
        **end if**
        Every $C$ steps update the target $Q$ network $\psi^- \leftarrow \psi$
        $s_t \leftarrow s_{t+1}$
    **end for**
**end for**

---

## 4. Experiments

We test the performance of algorithms on a range of scenarios with different scales that are designed based on the NASim [18] project. The experiments are divided into two parts: the first is to use a standard scenario as a benchmark to test the performance of different algorithms, and the second is to investigate the scalability using different scale scenarios. The experiment uses PyTorch as the framework of algorithms and is conducted on the following experimental environment: NVIDIA Geforce RTX3090 GPU, Intel Xeon Gold 6248R CPU, and 64GB RAM .

### 4.1. Network Scenarios

NASim is an open-source research platform that provides a series of simulated and abstract network scenarios for testing autonomous pentesting agents using the RL algorithms. It also offers a scenario generator that works as an option to generate networks automatically, allowing for rapid testing of the agent on different size of networks. The benchmark scenarios and the scenario generator are based on prior works [3,30].

The benchmark scenario for testing different algorithms is shown in Figure 3. To make it more complex and realistic, a honeypot is added to one of the benchmark scenarios in NASim, which is coming from practical commercial experience of [3]. This scenario has 7 subnets and 17 hosts, among which subnet 2 and 3 are internal networks and subnet 7 is a honeypot. The firewalls restrict the communication between subnets: only a small amount of service traffic is allowed through the firewalls. Each host can be configured with the following properties: address (defined by subnet number and machine number), OS (operating system), host value (measuring the importance of hosts), services (vulnerable software), and process (software that can be used to escalate privilege). Detailed configurations of scenario 1 are presented in Table 1 .

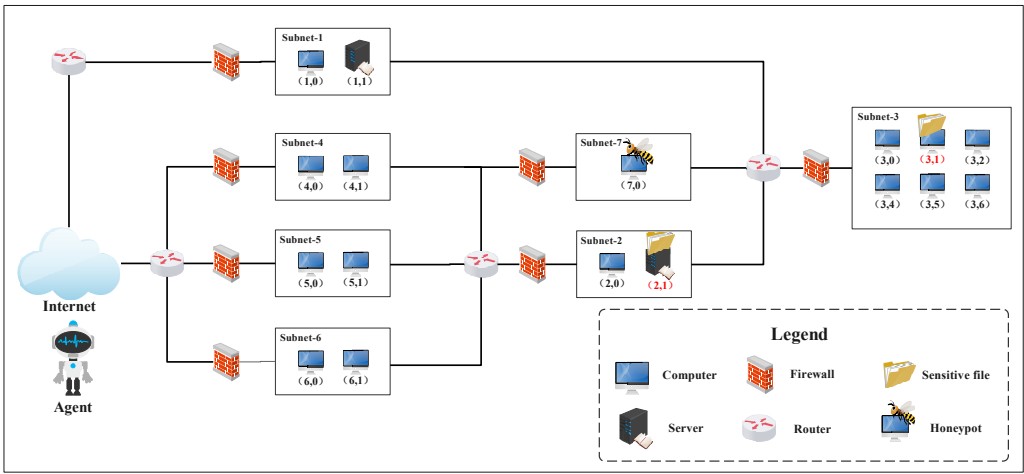

**Figure 3.** Structure of experimental scenario 1. Sensitive hosts are marked in red font.

The goal of the agent is to get root access to sensitive hosts by lateral movement in the network. Operations of the agent cover scan (gathering information of hosts), exploit (EXP, using vulnerable services to get root or user access to hosts ), and privilege escalation (PE, using processes to elevate the user access to root access). As the reward function is defined as Equation (5), the aim of the agent is to attack the most valuable hosts and get root access with the least costly operations as much as possible. Thus, the value of sensitive hosts is a large positive number, while that of the honeypot host is a negative number. In addition, each operation has an associated cost that is the comprehensive quantification of time, skills, and money costs. Furthermore, in order to simulate the uncertainty of attacks in the real world, every operation has a probability of success (POS). Here, we assume that the POS of the scan and PE operations are 1 and the others are determined by the CVSS (common vulnerability scoring system) score [31], which uses 0.2, 0.6, 0.8 to

simply determine the POS of a particular vulnerability. Whether an action can be executed successfully depends not only on the POS but also on the network topology, configuration of hosts, and the firewall. The detailed operations of agent in scenario 1 are shown in Table 2.

**Table 1.** Configurations of each host in scenario 1. Five vulnerable services, three processes, and two operating systems running in computers are chosen to configure the hosts and also define their points of vulnerabilities. Hosts with value of 0, 100, and −100 represent the common hosts, sensitive hosts, and the honeypot host, respectively.

| Address | OS | Value | Service | Process |
|---|---|---|---|---|
| (1,0) | Linux | 0 | SSH | Tomcat |
| (1,1) | Linux | 0 | SSH | Tomcat |
| (2,0) | Windows | 100 | SMTP | / |
| (2,1) | Windows | 0 | SMTP | Schtask |
| (3,0) | Linux | 0 | SSH | Tomcat |
| (3,1) | Linux | 0 | SSH, HTTP | / |
| (3,2) | Linux | 0 | SSH | / |
| (3,3) | Linux | 0 | SSH | / |
| (3,4) | Linux | 100 | SSH | Tomcat |
| (4,0) | Windows | 0 | FTP | Daclsvc |
| (4,1) | Windows | 0 | FTP | Daclsvc |
| (5,0) | Windows | 0 | FTP | Daclsvc, Schtask |
| (5,1) | Windows | 0 | FTP, HTTP | / |
| (6,0) | Linux | 0 | SSH | Tomcat |
| (6,1) | Windows | 0 | SSH, SMB | / |
| (7,0) | Windows | −100 | ALL | Daclsvc, Schtask |

**Table 2.** Operation list of the agent in experimental scenario 1. Eight EXP and PE operations are associated with the services and processes running in computers. Four kinds of scan operations can be used for an agent to obtain knowledge of the hosts' configuration.

| Operation | OS | Cost | POS | Access |
|---|---|---|---|---|
| SSH-EXP | Linux | 3 | 0.8 | USER |
| FTP-EXP | Windows | 1 | 0.5 | ROOT |
| HTTP-EXP | Linux | 2 | 0.8 | USER |
| SMB-EXP | Windows | 2 | 0.2 | ROOT |
| SMTP-EXP | Windows | 3 | 0.5 | USER |
| Tomcat-PE | Linux | 1 | 1 | ROOT |
| Daclsvc-PE | Windows | 1 | 1 | ROOT |
| Schtask-PE | Windows | 1 | 1 | ROOT |
| Subnet-Scan | / | 1 | 1 | / |
| OS-Scan | / | 1 | 1 | / |
| Service-Scan | / | 1 | 1 | / |
| Process-Scan | / | 1 | 1 | / |

In order to investigate the scaling performance, scenario generator is used to generate scenarios 2, 3, and 4 based on the given parameters of numbers of hosts and services in the network. The distribution of configurations of hosts is generated using a nested Dirichlet process to make the configurations of hosts across the network correlate. Rules of firewalls and POS of each service are generated randomly. Table 3 provides details of the generated scenarios used in the experiments. The number of sensitive hosts are fixed to 2 so that rewards become more and more sparse as the network size increases. The size of network deeply affects the action space of the agent. The number of operations is calcuated as *Services* + *Processes* + 4 where 4 means 4 types of scan operations.

In the initial state, the agent is located on the Internet and has no knowledge of the global information of the target network. The agent needs to select a series of proper actions to move laterally according to the topology information and host configuration information

obtained from the scan operation. An action refers to a certain operation taken against a certain host, and the ordered sequence of actions is the attack path. The agent learns the optimal attack path through exploration in each episode with limited training steps. The conditions for the end of each episode are: obtaining the root access of all sensitive hosts, the number of training steps reaching the set maximum, or attacking the honeypot hosts.

**Table 3.** Experimental network scenarios list.

| Scenario | Hosts | Sensitive | Subnets | Services | Processes | Operations | Honeypots |
|----------|-------|-----------|---------|----------|-----------|------------|-----------|
| Scenario 1 | 17 | 2 | 7 | 5 | 3 | 12 | yes |
| Scenario 2 | 80 | 2 | 17 | 10 | 4 | 18 | no |
| Scenario 3 | 100 | 2 | 21 | 10 | 4 | 18 | no |
| Scenario 4 | 150 | 2 | 31 | 10 | 4 | 18 | no |

*4.2. Results and Discussion*

The first part of our experiments is comparing the convergence performance of NDSPI-DQN, decoupling NDSPI-DQN, and the DQN algorithm under the same hyperparameter values (shown in Table 4). DQN is the baseline, as it has been widely used in many prior research studies [14–16]. Three algorithms are tested using scenario 1, and we compare the performance of them by looking at two metrics: the mean rewards over training steps during training episodes and the number of steps that the agent used in each episode.

As shown in Figures 4 and 5, at the start of training, the mean reward obtained by the agent is low and the mean episode steps reach the set maximum, indicating that the agent tends to use random policy to choose actions. As the training progresses, the agent gradually learns to use as few steps as possible to obtain the maximum rewards. Finally, both NDSPI-DQN and its decoupling version can converge on the approximate optimal value within limited episodes (~600 episodes for the decoupling version and ~1000 for the NDSPI-DQN) while DQN fails to converge during the training process.

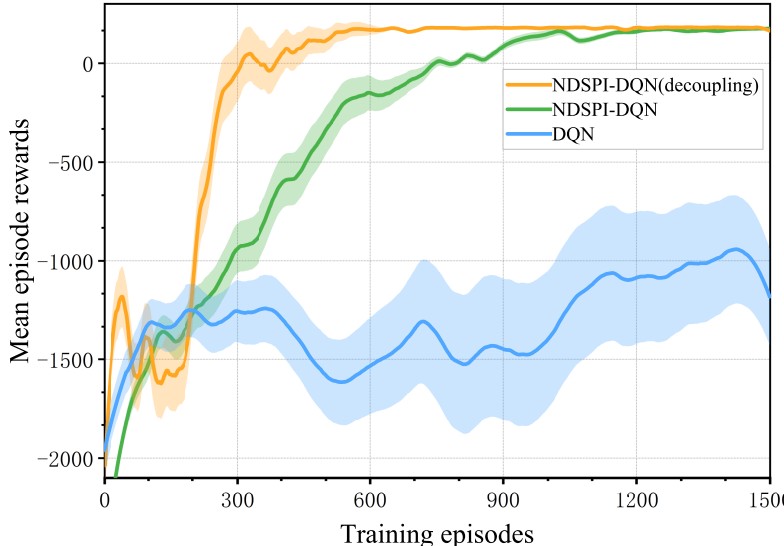

**Figure 4.** Mean episode rewards versus training episodes for different algorithms in scenario 1.

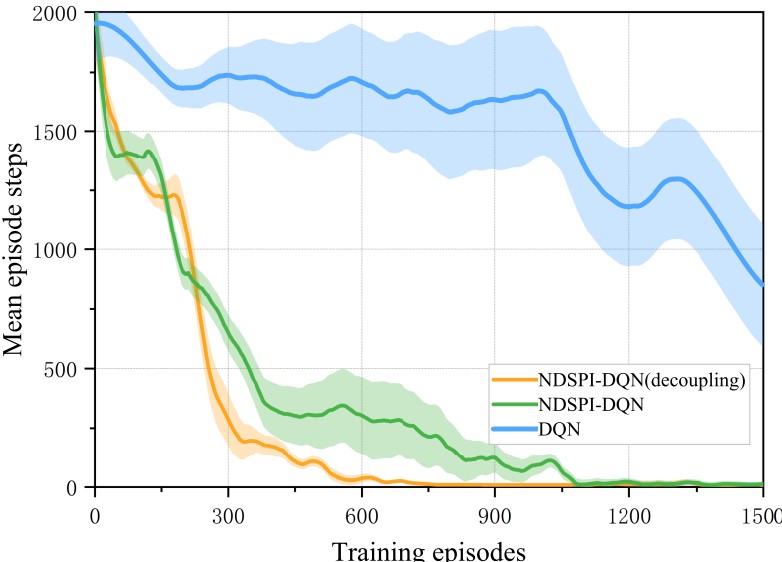

**Figure 5.** Mean episode steps versus training episodes for different algorithms in scenario 1.

The second part of experiments is investigating the scaling performance of our improved algorithms in different generated scenarios. The general procedure is to use scenario 2, 3 and 4, which are different in network sizes, test the NDSPI-DQN algorithm and its decoupling version with the same hyperparameter values (Table 4), and measure the convergence performance by looking at the mean rewards across the limited training episodes (Figure 6). DQN is ignored for its poor performance in the first part of experiments.

Figure 6 shows that the performance of the NDSPI-DQN algorithm is significantly affected by the scale of the network: when the network size increases to 120 hosts (Figure 6b) or more, the algorithm becomes difficult to converge. By contrast, the decoupling NDSPI-DQN algorithm can stably converge within limited episodes for all test network scenarios (∼200 episodes for scenario 2, ∼300 episodes for scenario 3, and ∼400 episodes for scenario 4). The size of the network hardly affects the convergence performance of the algorithm, which means that the algorithm has better robustness in large-scale network scenarios.

The experimental results indicate that the decoupling NDSPI-DQN converges faster and has the best performance in large-scale scenarios. The major reasons are:

1.  Integration of improvements promotes efficient exploration so as to alleviate the sparse reward problem. Rewards drive the agent's learning and act as supervisory signals. When the network scale increases, the reward becomes sparse, which means that the agent cannot obtain positive rewards in most of the training steps. By integrating a series of improved methods, we have greatly improved the exploration ability of the DQN algorithm, so that the agent can learn the optimal policy efficiently within a limited number of training steps per episode.
2.  The reduction of action space reduces the cost of trial and error. As we mentioned previously, an action is an attack vector that is defined as $< h, o >$, and the action space of the agent is up to $O(M \times N)$. For example, assuming that the agent in scenario 1 has compromised the host (4,1), then it can take action $< (2, 0), scan >$ to scan the host (2,0). Since there are 17 hosts, and for each host, the agent can take 12 types of executable operations in scenario 1, the total action space reaches $17 \times 12 = 204$. As the network scale increases, the huge action space makes it difficult for the agent to select the proper action. We use two streams to separately estimate the target host and executable operations according to the environment state, and then combine them into an attack vector to act on the environment, so that the agent's action space is reduced to $O(M + N)$, thus speeding up the convergence speed in large-scale network scenarios.

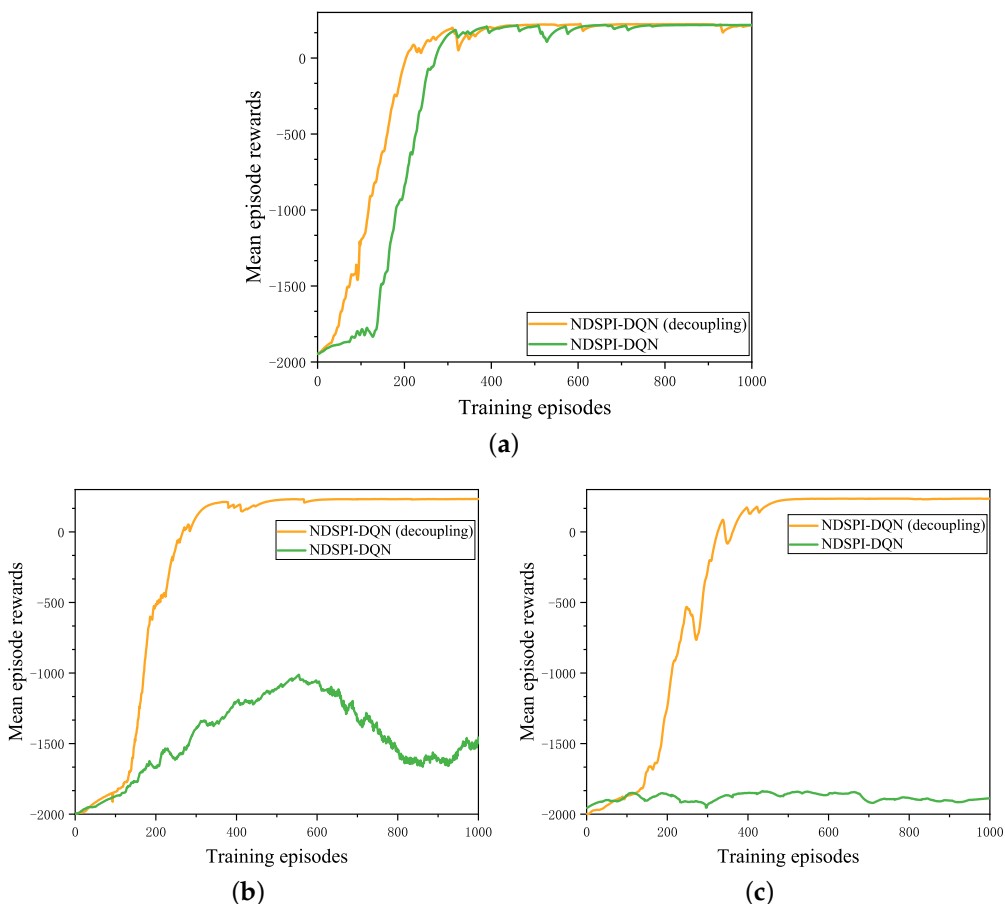

**Figure 6.** Mean rewards versus training episodes in different scenarios. (**a**–**c**) show results for scenario 2 (80 hosts), scenario 3 (120 hosts) and scenario 4 (150 hosts), respectively.

Additionally, the scale of the network is not the only factor that affects the performance of the algorithms. By comparing the convergence performance of the algorithms in scenario 1 (Figure 4) and scenario 2 (Figure 6a), we can find that although the scale of scenario 2 is larger than scenario 1, the algorithms can converge faster. The reason is that except for sensitive hosts with positive rewards, there are also honeypot hosts with negative rewards in scenario 1. Once the honeypot host is compromised by the agent, the training episode will end (in the real world, if the hackers attack the honeypots, the pentesting task will be exposed and failed), making it more difficult for the agent to learn the optimal policy. This is also a manifestation of the effect of the reward value on the performance of the algorithms, and our method can be applied to these complex network scenarios well.

**Table 4.** List of hyperparameter values of algorithms.

| Hyperparameter | Value |
|---|---|
| Max steps per episode | 2000 |
| Learning rate | 0.0001 |
| Batch size | 64 |
| Discount factor, $\gamma$ | 0.9 |
| Hidden layer size | 256 |
| Replay memory size | 300,000 |
| Target network update frequency, $C$ | 1000 |

## 5. Conclusions and Future Work

In this paper, we model pentesting as an MDP problem and apply the deep reinforcement learning algorithm to solve it. By summarizing previous work, we propose the algorithm named NDSPI-DQN, which has better robustness for the autonomous pretesting task in large-scale network scenarios. Specifically, we reasonably select a series of DQN improvement methods and integrate them into the DQN algorithm to improve the exploration ability, so as to better solve the sparse reward problem. In addition, by decoupling the attack vector, we significantly reduce the action space of the agent, thereby reducing the trial and error cost in the agent's exploration process. We construct simulation network scenarios of different scales to train agents, and experiments show that our proposed algorithms have better convergence performance in large-scale and complex network scenarios.

We provide scalable and robust RL algorithms that can be used to train PT agents. However, at present, autonomous penetration testing using reinforcement learning is still in the stage of simulation experiments, for the reason that modeling the actual network traffic is hard and the trial-and-error approach in the real world is costly. This is a significant limitation in our method that needs to be addressed in future contributions. The application of emulation technologies and virtualization technologies may be a future direction worth exploring. In addition, using more advanced algorithms such as multiagent RL or hierarchical RL to improve the performance of algorithms for more cyberattack tasks from the MITRE matrix is also an interesting and meaningful direction for future work.

**Author Contributions:** Conceptualization, S.Z. and J.L.; methodology, S.Z.; software, S.Z.; validation, S.Z., J.L., D.H. and Y.Z.; formal analysis, J.L. and D.H.; investigation, X.Z. and Y.Z.; resources, S.Z. and X.Z.; data curation, S.Z.; writing—original draft preparation, S.Z.; writing—review and editing, X.Z., Y.Z. and D.H.; visualization, Y.Z.; supervision, J.L.; project administration, S.Z. and J.L. All authors have read and agreed to the published version of the manuscript.

**Funding:** This research received no external funding.

**Institutional Review Board Statement:** Not applicable.

**Informed Consent Statement:** Not applicable.

**Data Availability Statement:** The data presented in this study are available on request from the corresponding author. The data are not publicly available due to privacy.

**Conflicts of Interest:** The authors declare no conflict of interest.

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
