# Peer review of "Autonomous Penetration Testing Based on Improved Deep Q-Network"

_applsci, doi:10.3390/app11198823_

Round 1

Reviewer 1 Report

The structure of this article is very well designed - descriptive, and in the same time - compact. The chosen models are innovative and new in test. For further research, try to enlarge the network implying more hosts.

The main question in this research is in the field of artificial intelligence in cybersecurity, more specifically – test and verifiaction of an improved algorithm called NDSPI-DQN. It is relevant to cybersecurity automation systems and will be interesting to researchers adopting AI for security. This paper is original in the level the reviewer has never met the proposed algorithm and network topology for testing it before. The paper adds new knowledge and test results for this improved NDSPI-DQN algorithm, which may act as a future implementation in software instruments for network penetration testings. The written text is correct, clear and easy to read. Ommited are bulky scientific explanations.

The conclusion is made by comparing previous algorithm results with the newly proposed one. Quantative correlation is the proof of consistency.

Author Response

Thank you for your approval of the article. We will work on further research.

Reviewer 2 Report

The research described in the manuscript significantly adds value to the research area.

However, none of the code, models, datasets have been made public significantly reducing the impact of the research done. It also raises concerns about the reproducibility of the approach. 

Author Response

Thanks to you for your good comments. Our code will be open-source soon.

Reviewer 3 Report

The topic is relevant, the approach shows a new perspective on the idea of testing network security

Author Response

(The authors gave the same response as above.)
